# A Controlled System for Parahydrogen Hyperpolarization Experiments

**DOI:** 10.3390/molecules30214299

**Published:** 2025-11-05

**Authors:** Lorenzo Franco, Federico Floreani, Salvatore Mamone, Ahmed Mohammed Faramawy, Marco Ruzzi, Cristina Tubaro, Gabriele Stevanato

**Affiliations:** 1Dipartimento di Scienze Chimiche, Università di Padova, Via Marzolo 1, 35131 Padova, Italy; lorenzo.franco@unipd.it (L.F.); federico.floreani@studenti.unipd.it (F.F.); ahmed.faramawy@unipd.it (A.M.F.); marco.ruzzi@unipd.it (M.R.); cristina.tubaro@unipd.it (C.T.); 2Department of MESVA (Life, Health & Environmental Sciences), Università dell’Aquila, Via Vetoio SNC, Localita’ Coppito, 67100 L’Aquila, Italy; salvatore.mamone@univaq.it

**Keywords:** PHIP, SABRE, hyperpolarization, bubbling setup

## Abstract

Parahydrogen-induced hyperpolarization (PHIP), introduced nearly four decades ago, provides an elegant solution to one of the fundamental limitations of nuclear magnetic resonance (NMR)—its notoriously low sensitivity. By converting the spin order of parahydrogen into nuclear spin polarization, NMR signals can be boosted by several orders of magnitude. Here we present a portable, compact, and cost-effective setup that brings PHIP and Signal Amplification by Reversible Exchange (SABRE) experiments within easy reach, operating seamlessly across ultra-low-field (0–10 μT) and high-field (>1 T) conditions at 50% parahydrogen enrichment. The system provides precise control over bubbling pressure, temperature, and gas flow, enabling systematic studies of how these parameters shape hyperpolarization performance. Using the benchmark Chloro(1,5-cyclooctadiene)[1,3-bis(2,4,6-trimethylphenyl)imidazole-2-ylidene]iridium(I) (Ir–IMes) catalyst, we explore the catalyst activation time and response to parahydrogen flow and pressure. Polarization transfer experiments from hydrides to [1-^13^C]pyruvate leading to the estimation of heteronuclear J-couplings are also presented. We further demonstrate the use of Chloro(1,5-cyclooctadiene)[1,3-bis(2,6-diisopropylphenyl)imidazolidin-2-ylidene]iridium(I) (Ir–SIPr), a recently introduced catalyst that can also be used for pyruvate hyperpolarization. The proposed design is robust, reproducible, and easy to implement in any laboratory, widening the route to explore and expand the capabilities of parahydrogen-based hyperpolarization.

## 1. Introduction

Nuclear magnetic resonance (NMR) and magnetic resonance imaging (MRI) are valuable techniques used throughout chemistry, physics, and medicine [1,2,3]. Yet their impact is inherently limited by low sensitivity and poor signal-to-noise ratios (SNRs). The nuclear spin polarization, representing the fractional imbalance between energy levels responsible for the NMR signal, follows a simple expression, proportional to ℏγB_0_/2kT, where ℏ is the reduced Planck constant, γ is the gyromagnetic ratio, B_0_ is the magnetic field, k is the Boltzmann constant, and T is the temperature [4]. Conventional strategies to overcome this limitation rely on higher fields and/or lower temperatures. Brute-force nuclear hyperpolarization follows this route [5,6,7,8]. Reaching very high fields, however, is technically demanding. Commercial NMR magnets based on superconducting alloys such as NbTi or Nb_3_Sn require cooling with liquid helium. The largest commercially available systems operate at ~28 T, weigh several tons, and cost millions of euros, yet still cannot break the fundamental scaling granting only up to about 1 proton spin over 10^4^ to contribute to the NMR signal. On the other side, cryoprobes and microcoils can be used to reduce noise on the detection side [9,10].

Hyperpolarization provides an alternative [11,12,13,14]. Several distinct methods have been developed, depending on the source of polarization (i.e., the signal) to be transferred to the nuclear spins: from electron spins, as in Dynamic Nuclear Polarization (DNP) [12,15,16,17,18,19,20,21]; from parahydrogen, as in Parahydrogen-Induced Polarization (PHIP) and Signal Amplification by Reversible Exchange (SABRE) [22,23,24,25,26,27]; from noble gases, as in Xe- and He-polarization [28]; or from quantum tunneling effects, as in Quantum Rotor-Induced Polarization [29,30]. DNP remains the most widely adopted method, capable of yielding signal enhancements of several hundredfold—either in solids under magic-angle spinning at cryogenic temperatures or in liquids via stochastic electron–nuclear interactions (Overhauser effect) [18,31,32,33,34]. Dissolution DNP (dDNP) extends this signal amplification by several thousand-fold to solution-state NMR by polarizing frozen samples at ~1.2 K and rapidly dissolving them, enabling enhanced detection of metabolic probes such as pyruvate [3,16,19,35,36]. Despite its impact, DNP remains complex, expensive, and difficult to maintain.

Parahydrogen-based methods offer a simpler path [22,23]. PHIP and its variant PHIP Side-Arm Hydrogenation (PHIP-SAH) [37,38] can match the performance of dDNP for many targets, while SABRE [26,27] allows repeated experiments on the same sample at a fraction of the cost. These techniques can hyperpolarize nuclei in seconds to minutes, without cryogenics or high-power microwaves. Their efficiency, however, depends critically on controlling key variables: magnetic field, parahydrogen enrichment, bubbling time, pressure, flow rate, hydrogen and substrate solubility and chemical kinetics in the used solvent, and the symmetry of the molecular system [39,40,41,42].

In this work, we present an easily implementable parahydrogen and bubbling apparatus that provides control over pressure, flow rate, and temperature during parahydrogen-based hyperpolarization experiments.

Several systems for parahydrogen at different enrichment levels have been reported in the literature [41,43,44,45,46,47]. Some use partially 3D-printed materials [46] and some operate at pressures around 490 psi (≈33 bar) and reach enrichments above 48%, up to nearly 100%, with designs aimed at clinical applications [43,46,47,48,49]. Commercial solutions are also available from Bruker (e.g., the BPHG90, delivering ≈87% parahydrogen at a cost above EUR 100 000; Bruker Switzerland AG, Industriestrasse 26 8117 Fällanden, Switzerland (CH), Hyperspin Scientific UG (http://www.hyperspin.biz/, accessed 1 November 2025, Mathystraße 19 76133 Karlsruhe, Germany (DE)), and Xeus Technologies (https://www.xeus-technologies.com/, accessed 1 November 2025, Schediasmou Lakatamia, Nicosia 2326, Cyprus (CY)), which additionally offer a complete parahydrogen distribution system.

Our goal here is not to introduce an alternative method but rather to demonstrate that a wide range of experiments—with full control over key experimental parameters—can be performed in almost any research environment, including for educational purposes, using relatively modest resources.

In many laboratories, parahydrogen experiments are still performed by manually shaking a *J*-Young NMR tube in the fringe field of a magnet, followed by detection with a high magnetic field [50,51,52,53]. While this approach can be effective, it raises questions about how well-controlled such experiments truly can be. Conversely, the full automation of parahydrogen bubbling may appear as a technical barrier—one that this work aims to overcome. We describe in detail the construction of a custom switch box that can be operated manually for parahydrogen experiments, while also enabling fully automated high-field operation through the TTL trigger signals available from most NMR spectrometer consoles. Practical examples of implementation are also provided, including pulse sequences used on Bruker instruments.

The proposed system is then employed to evaluate the performance of a recently developed SABRE catalyst for sodium [1-^13^C]pyruvate experiments, Chloro(1,5-cyclooctadiene)[1,3-bis(2,6-diisopropylphenyl)imidazolidin-2-ylidene]iridium(I) (Ir–SIPr), using a 1.4 T benchtop NMR spectrometer (XPulse, Oxford Instruments Magnetic Resonance, Tubney Woods, Abingdon Oxfordshire, OX13 5QX (UK)), demonstrating ^13^C hyperpolarization under controlled bubbling conditions, although the sample transfer to the detection field is still performed manually. The same setup is used to perform fully automated high-field experiments (9.4 T) with the well-known Ir–IMes Chloro(1,5-cyclooctadiene)[1,3-bis(mesityl)imidazolin-2-ylidene]iridium(I) [54,55] to accurately monitor the catalyst activation time by following the hydride ^1^H signals to conduct polarization transfer experiments from parahydrogen protons to the ^13^C nucleus of [1-^13^C]pyruvate and to study the dependence of the ^13^C signal intensity on hydrogen flow at 6 bar pressure. Overall, the system presented here—operating even with a 50% parahydrogen fraction—enables a broad range of experiments with both the low magnetic fields typical of benchtop systems and high magnetic fields, thereby significantly widening the applicability of parahydrogen-based hyperpolarization methods.

## 2. Description of the System

### 2.1. Parahydrogen Generator

In many laboratories, the hydrogen source is typically represented by high-pressure gas cylinders (≈200 bar), which are commercially available at relatively low cost. In our case, however, considerations related to logistics, safety regulations, and the desire to design a compact and easily transportable setup—allowing parahydrogen experiments to be performed on different NMR instruments—led us to adopt a different solution. A detailed list of components and estimated costs (EUR < 13 k, excluding the hydrogen generator) is provided in Appendix A.

Hydrogen gas is provided by a commercial hydrogen generator (AD-600 Cinel SRL- Italy) (Figure 1b(1)). This unit produces high-purity (>99.99999%) H_2_ gas via a PEM (Proton Exchange Membrane) electrolytic H_2_ generator using deionized water (low conductivity (<1 µS/cm)). In our case the H_2_ gas flow is 600 mL/min and it is produced at a user-controllable pressure of up to 10 bars. In our experiments, the hydrogen pressure was set to 7.5 bar. The gas produced by the generator was delivered through a 1/8″ Teflon tube, which was subsequently coupled to a 6 mm o.d. copper line. A S-Lok manual ball valve (S-Lok SBV1-3B-S-6M, Mobile, AL, USA) regulated the gas flow into the system (Figure 1d(2)), while a first-pressure gauge (Figure 1d(4)) monitored the internal pressure (Swagelok PGI-63B-BG16-CASX, Solon, OH, USA). In parallel with the gas inlet, a second 6 mm o.d. outlet copper line was connected via a second manual valve (S-Lok SBV1-S-6M) (Figure 1d(3)) to a diaphragm pump (Vacuubrand Model MZ 1C, Wertheim, Germany) (Figure 1b(12)), which was activated only at the end of the experimental session to purge residual H_2_ gas from the circuit. To prevent system over-pressurization when residual gas in the line warms from liquid-nitrogen to room temperature, a safety pressure relief valve (IMI Norgren 1002/BR008, Lichfield, UK) (Figure 1d(9)) with an automatic release at 14 bar was installed.

Hydrogen gas enters a soft annealed copper spiral (Figure 1b(10)) into which approximately 21 g of iron(III) oxide powder (Merck, 371254-50G, 30–50 mesh, CAS 20344-49-4, Darmstadt, Germany) had been manually introduced and packed prior to coiling the tube.

The spiral consists of 10.5 turns with an outer diameter of 65 mm. On both ends of the copper coil, compressed cotton wool plugs (approximately 20 mm in length) were inserted to retain the catalyst within the tubing.

Before loading, the iron(III) oxide powder was washed to remove fine particles that could otherwise be transported downstream and promote back-conversion of parahydrogen to orthohydrogen. Specifically, the powder was poured into a glass funnel fitted with a Whatman No. 1 filter paper (retention ~11 µm) and washed three times with n-Hexane until the filtrate appeared clear. The powder was then dried overnight at 60 °C in an air oven.

The resulting copper spiral filled with the cleaned iron(III) oxide catalyst is immersed in a 2 L Dewar (Isotherm DSS 2000, Karlsruhe, Germany) (Figure 1b(11)) containing liquid nitrogen at 1 bar and 77 K, where it acts as the parahydrogen conversion unit. At both the inlet and outlet of the copper spiral, two 50 µm in-line filters (Parker M6A-F4L-50-SS, Cząstków Polski, Poland) (Figure 1d(5)) are installed to further minimize the chances of iron particles migrating into the warm horizontal section of the parahydrogen generator shown in Figure 1d. A pressure gauge, placed after an S-Lok manual ball valve (Figure 1d(6)), monitors the gas pressure downstream of the iron(III) oxide-filled spiral (Figure 1d(7)). A mass-flow controller (Sierra Instruments C100, connected to a 24 VDC power supply, Monterey, CA, USA) (Figure 1d(8)) is used to regulate the flow of gas entering the distribution system. During Signal Amplification by Reversible Exchange–Shield Enables Alignment Transfer to Heteronuclei (SABRE-SHEATH [39,40]) catalyst activation, a low flow rate of approximately 20–30 sccm, adjustable and displayed on the mass-flow controller, is maintained for 20 min to prevent excessive solvent evaporation. During hyperpolarization experiments, the flow is typically set to ~80 sccm and can be increased up to 150 sccm.

### 2.2. The pH_2_ Distribution System

The distribution system is the part of the apparatus responsible for delivering the pH_2_ gas from the parahydrogen generator to the NMR tube (see Figure 1c). The 7.5 bar pH_2_ pressure after the mass-flow controller is reduced to 6 bars by the pressure regulator valve (Festo MS2-LR-QS6-D6-AR-BAR-B, Esslingen am Neckar, Germany) (Figure 1c(1)). However, we also successfully tested a final pressure of 8 bar at the NMR tube during the test experiments. In order to deliver the pH_2_ gas in a controlled way to the NMR tube, three valves are necessary: an inlet, an exhaust, and a bypass valve. Although we used manual valves during the initial testing session, in the actual setup they have been all replaced with solenoid valves (SMC SMVDW20HA) (Figure 1c(2,6,7)) that can be remotely controlled as described below. They require a 24 V DC power supply unit (shown in the expansion of Figure 1c(4)). The inlet and exhaust lines, schematically sketched in (Figure 1a), guide the pH_2_ through the NMR tube, creating the desired bubbling when the bypass valve is closed. When it is open, the bypass valve connects the inlet and outlet and the bubbling stops while keeping the system under pressure (see Figure 1a). The desired pressurization of the system is guaranteed by a back-pressure regulator (Rometec SRL—back-pressure regulator, ¼” NPTF, body: 316 SS, seat retainer: PEEK, diaphragm: 316 SS, o-ring: Viton, 0–17 bar) equipped with an in-line pressure gauge (Figure 1c(3)) that is manually adjusted in order to keep the pressure slightly below the value dictated by the pressure regulator valve (Figure 1c(1)).

The 4 mm o.d. flexible tubing used for the inlet and outlet gas lines was connected via a push-in Y-adapter (two 4 mm and one 6 mm ports, shown in Figure 1a(1)). The 6 mm port was fitted with a Teflon tube terminated with a nickel-plated brass compression fitting, allowing easy disconnection when required (Figure 1a(2)). The nickel-plated brass compression fittings were used to ensure a secure and leak-tight connection between the 6 mm o.d. Teflon lines (with tubing in Figure 1c) and the 4 mm o.d. flexible tubing used to deliver the gas to the NMR tube. The nickel-plated brass connectors (Figure 1a(2)) were also utilized in the experiments reported below at magnetic fields of 9.4 T, displaying no observable magnetic behavior. We employed a portable H_2_ gas detector to monitor possible leaks in the apparatus (see Appendix A, RS COMPONENTS—RS GD38).

To minimize field inhomogeneities in the NMR detection region, a 250 µm i.d. quartz capillary (Molex, Part Number: 1068150026) was glued to a 1/16″ o.d. flexible Teflon tube, which was then inserted and bonded to the inner wall of the 4 mm o.d. tube using a two-component epoxy adhesive.

The core of the distribution system is represented by the control unit in Figure 1c(4) (expanded in Figure 1e,f). It operates with AC 110/220 V, 50 Hz mains input, converted to 24 V DC (250 W) through a regulated power supply, as detailed in the schematic diagram in Figure 2. The DC output feeds a four-pole double-throw (4PDT) MAN/AUTO switch, allowing the operator to select between manual and automatic operating modes. In manual mode, four illuminated pushbuttons (red, green, blue, and white—see also Figure 1f) provide direct activation of each output channel. Each button lights up when pressed, indicating that the corresponding circuit is energized. In automatic mode, control is transferred to a four-channel relay board (24 V DC, opto-isolated) equipped with electromechanical relays rated for up to 10 A at 250 V AC or 30 V DC. Each relay drives a 24 V DC solenoid valve that regulates gas flow within the system. The optocoupler interface ensures electrical isolation between the control logic and the load lines. When the system operates automatically, as in the 9.4 T experiments reported below, a series of LEDs switch on to indicate which valves are active, thus providing consistent visual feedback across both operating modes.

All components are housed within an aluminum enclosure (approximately 200 × 150 × 80 mm) with front-panel indicators, a MAN/AUTO switch, and connectors for external wiring. The design enables straightforward transition between manual testing and automated operation, ensuring robust and reproducible control of the solenoid valves under low-voltage (24 V DC) conditions. Although the solenoid valves heat up if kept active for extended periods (e.g., bubbling times longer than 1 min), potentially leading to malfunctions in their opening and closing mechanisms, no such issues were observed during our experiments, even for bubbling durations of up to 10 min and continuous operation over several hours. In automatic mode, the solenoid valves are automatically deactivated at the end of each bubbling cycle.

### 2.3. The Temperature Control Unit and μ-Metal for SABRE-SHEATH Experiments

The temperature control unit consists of a Wilmad suprasil VT dewar (see Figure 3a). This insert (outer diameter: 10.5 mm; lower section length: 242 mm; upper section length: 72 mm) was constructed from high-purity Suprasil^®^ fused silica (quartz glass). It was designed as a non-silvered, unslotted insert, offering excellent optical purity, structural stability, and very low thermal expansion over a broad temperature range (ranging from ~120 K up to ~600 K in standard use). In our experimental setup (see the scheme in Figure 3b), the Suprasil VT Dewar is connected to a liquid-nitrogen reservoir via an evaporator. At the bottom of the suprasil dewar, a thermocouple measures the temperature (T probe in Figure 3a). The temperature probe is mounted so that it is in contact with the outer wall of the NMR tube to monitor the temperature of the gas directly. That sensor is interfaced with an external control unit (Bruker BVT2000) which dynamically adjusts the flow of N_2_ gas from the evaporator to regulate the temperature. Through this feedback scheme, we observed temperature stability better than ±0.1 K over periods of several hours. We used this system because it was already available in our laboratory and had been employed for low-temperature EPR measurements. However, this system is now discontinued. Vendors such as Bruker (Bruker Switzerland AG, Industriestrasse 26 8117 Fällanden, Switzerland (CH)), Oxford Instruments (Oxford Instruments Magnetic Resonance, Tubney Woods, Abingdon Oxfordshire, OX13 5QX (UK)), and others offer integrated solutions that can be purchased if needed.

To perform SABRE-SHEATH hyperpolarization experiments on heteronuclei (e.g., ^13^C, ^15^N, etc.), the magnetic field during bubbling must be accurately adjusted to sub-microtesla levels, well below the Earth’s magnetic field. In practice, this can be achieved by shielding the bubbling region from external magnetic fields using a metal enclosure, inside which a solenoid coil (Figure 3a) is installed and driven with a DC current to generate the desired longitudinal magnetic field. For completeness, we note that for ^1^H SABRE-SHEATH experiments the field requirements (usually around 6 mT) are less stringent and might not require a metal shield. In the proposed setup, concentric with the μ-metal shield are the solenoid coil, which generates the desired magnetic field, and the Wilmad dewar. The Wilmad dewar is placed at the center of the µ-metal shield (Sas Ateliers Soudupin 4 Rue du Pharle 77130 Montereau Fault Yonne (FR), Zero gauss chamber, model: ZG-100-300-3-0, with three layers and a μ-metal thickness of 1 mm; internal diameter: 100 mm, length: 300 mm), which contains a solenoidal coil (364 turns of 1 mm copper wire wound in two layers for a total length of 18.5 cm) coupled via a 10 kΩ resistance to a DC power supply (model: Keysight E36231A) (Figure 3a) to generate the requested magnetic field for hyperpolarization experiments. Here we add that a recent study demonstrated a low-cost approach to parahydrogen-based hyperpolarization using custom solenoid magnets [57]. The Wilmad suprasil VT dewar is connected to a liquid nitrogen tank via an evaporator line. The sub-microtesla magnetic fields necessary for SABRE-SHEATH experiments were measured via a flux-gate magnetometer (our model, the Bartington MAG-03MC, Witney, UK, (±1000 µT), coupled to the MAG-Meter 2 features a typical resolution of ≈50 nT, while lower-range versions (±70 µT or ±100 µT) reach ≈ 5 nT). The calibration curve in Figure 3c was obtained, showing a good linearity between the applied voltage and the measured magnetic field.

### 2.4. Integration of TTL-Controlled Bubbling System for High-Field Parahydrogen Experiments

To perform high-field NMR experiments (i.e., 9.4 T for us), it is advantageous to employ a controlled bubbling system that can be directly operated and regulated by the pulse program initiating the NMR experiment itself. Here, we describe how the TTL output lines of the Bruker NMR console were used to control the state of the solenoid valves via the custom-built switch box shown in Figure 1c(4). In our setup, the Bruker Ascend 400 MHz spectrometer equipped with an Avance III NanoBay console provides four TTL output channels that can serve as trigger signals to control the solenoid valves. Since three valves were required for our system, only three of these lines were utilized. According to the Bruker manual (IPSO AQS Unit Technical Manual), several TTL lines—depending on the specific console configuration—can be assigned to transmit or receive trigger signals.

The implementation simply requires connecting the corresponding console pins to the switch box via BNC cables and editing the *Avance.incl* file to define the logical state (HIGH or LOW) of each valve during the pulse sequence execution. Once this configuration is established, the valve state can be controlled directly within the TopSpin pulse program by inserting the instruction TTL1 HIGH or TTL1 LOW for the valve associated with trigger 1 (and similarly for triggers 2 and 3). An example of a pulse sequence used for bubbling and detection is provided in Appendix B.

For Bruker NMR users, the first step is to consult the table listing the pin assignments for the RCP and control signals of the T-controller on the IPSO AQS (see Table 4 in the Avance III NanoBay IPSO AQS Unit Technical Manual). In our setup, the console allows sending an output signal through pin V6. Table 3 of the same manual identifies the V bundle and the corresponding V6 pin. The signal from this pin can be routed through a BNC connector cable and coupled to the output of the switch box illustrated in Figure 1c,e and Figure 2. To correctly use the V6 pin as a trigger output, the *Avance.incl* file must be modified. First, locate the *Avance.incl* file; then, refer to the appropriate table in the Avance III NanoBay IPSO AQS Unit Technical Manual to find the *setnmr* number corresponding to the desired pin and define the TTL line accordingly, as described in the Bruker manual. For instance, in our case the *Avance.incl* is located at C:\Bruker\TopSpin3.7.0\exp\stan\nmr\lists\pp, and the TTL line for pin V6 can be defined as follows:

/*trigger outputs 1*/;

#define TTL1_LOW setnmr4|14;

#define TTL1_HIGH setnmr4^14.

A similar syntax has been used for the other TTL lines.

## 3. Results and Discussion

As a demonstration of the versatility of the setup described above, we present a series of characterization experiments on the SABRE catalyst Ir–SIPr and gold-standard Ir–IMes, synthesized in our laboratory. The two complexes differ in terms of the coordinated NHC ligand: the Ir–SIPr complex features as the NHC ligand 1,3-bis(2,6-diisopropylphenyl)imidazolidin-2-ylidene, while Ir–IMes features 1,3-bis(2,4,6-trimethylphenyl)imidazolin-2-ylidene. The difference between the two NHCs, in addition to the substituents at the nitrogen atoms, lies in the backbone of the heterocycle, which can be saturated (SIPr) or not (IMes). The benchmark Ir–IMes remains, however, the best-performing NHC catalyst at the moment [56]. The sample was prepared as detailed in Section 4.

The SABRE mechanism for pyruvate follows the description given by Duckett and co-workers [58]. The NHC precatalyst (either Ir–SIPr or Ir–IMes) (**1**) is activated in methanol-d_4_ under a continuous parahydrogen flow (≈30 sccm) at 6 bar, with the activation times estimated below for Ir–IMes at 265 K. The co-ligand, DMSO, at an optimal 5 equivalent concentration with respect to the catalyst, is instrumental in the reversible binding pyruvate to the catalyst. In the absence of a co-ligand, no pyruvate hyperpolarization is observed. Upon binding, the resulting sulfoxide complexes [Ir(H)_2_(κ^2^-pyruvate)(DMSO)(NHC)] (**3**) (see Figure 4) act as the active polarization transfer species for ^13^C-pyruvate, while [IrCl(H)_2_(DMSO)_2_(NHC)] (**2**) (see Figure 4) mediates hydrogen exchange. Two coordination geometries can be identified for pyruvate binding: **3a** (axial) and **3b** (equatorial).

### Experiments at 1.4 T with a Benchtop NMR and at 9.4 T

Here we report SABRE-SHEATH experiments conducted at 1.4 T and complementary ^1^H and ^13^C measurements performed at 9.4 T using Ir–SIPr and Ir–IMes catalysts with the setup described above. The data for 1.4 T presented here concerning the variable temperature and variable pressure experiments in Figure 5 were previously reported in ref. [56] and are included here for completeness and clarity.

The level of ^13^C polarization, i.e., signal enhancement, for [1-^13^C]pyruvate was estimated for Ir–IMes to be approximately 3.5% in the experiments reported in ref. [56] and used in the present study. This value represents the average polarization obtained typically over at least ten repetitions using the same sample. Polarization levels depend on several factors, including—but not limited to—the lifetime of the catalyst, which is not addressed in detail here. Over the course of a day, signal enhancement tends to decrease gradually, likely due to catalyst degradation. Under comparable conditions of concentration, temperature, solvent, parahydrogen fraction, and bubbling time, the best polarization levels achieved with Ir–SIPr are approximately two-thirds of those obtained with Ir–IMes. Notably, using 100% parahydrogen increases the expected polarization by a factor of about three, reaching values of around 10% for Ir–IMes. Even higher levels, up to 20%, have recently been reported for [1-^13^C]pyruvate-d_3_ using SABRE-SLIC [59].

Differences between NHC–Ir catalysts can be rationalized in terms of their electronic and steric properties [60,61]. The Tolman Electronic Parameter (TEP) [62], derived from the ν(CO) stretching frequency in [Ni(CO)_3_L], and the Huynh Electronic Parameter (HEP) [63], obtained from the ^13^C chemical shift of the carbene carbon in *trans*-[PdBr_2_(*^i^*Pr_2_-bimy))(NHC)] (*^i^*Pr_2_-bimy = 1,3-diisopropylbenzimidazolin-2-ylidene), respectively, indicate the σ-donor/π-acceptor balance and the σ-donor strength of a given NHC ligand. The TEP values were 2050.7 cm^−1^ for IMes and 2052.2 cm^−1^ for SIPr in dichloromethane solution, and the HEP values were 177.2 ppm for IMes and 177.6 for SIPr measured in CDCl_3_ [60]. Additional probes such as the ^77^Se NMR shift of NHC–Se adducts provide complementary information on the electronic density at the metal center. The ^77^Se chemical shifts were 35 ppm for IMes and 181 ppm for SIPr in deuterated acetone [60]. Steric effects are usually quantified by the percent buried volume (%Vbur), which describes the fraction of the coordination sphere occupied by the ligand, and can vary depending on the different M-NHC complexes. Using the complex [IrCl(COD)(NHC)], the %Vbur values were 33.0 and 35.4 for IMes and SIPr, respectively, calculated for an M-NHC bond at 2.00 Å [61]. Taken together, these descriptors explain the distinct behavior of Ir–IMes and Ir–SIPr complexes, the latter NHC being both a stronger σ-donor and more sterically demanding ligand.

In SABRE-SHEATH, when bubbling is performed in the level-anticrossing (LAC) magnetic field (~0.33 μT), polarization is spontaneously transferred from parahydrogen to the *J*-coupled ^13^C nuclei. As a result, two main classes of signals can be observed: one corresponding to the free substrate (**4** in Figure 4) and up to two associated with substrate molecules bound to the catalyst (**3a** and **3b** in Figure 4).

Using Ir–SIPr, as shown in our work [56], we clearly detected, by ^13^C NMR, the free pyruvate signal at ~170 ppm (**4**) and the equatorially bound species at ~168.5 ppm (**3b**), while the axially bound pyruvate (**3a**) appeared only as a weak shoulder. The lack of a **3a** signal is related to the low polarization but reflects the sterically constrained dynamics imposed by the bulky diisopropylphenyl groups of the SIPr ligand, which hinder the formation or stabilization of the **3a** isomer. A similar trend was found for the unsaturated analogue, Ir–IPr, consistent with earlier observations linking ligand rigidity and donor strength to SABRE efficiency [54].

Figure 5a shows the temperature dependence of the ^13^C hyperpolarization for Ir–SIPr between 240 K and 305 K. Each sample was bubbled with parahydrogen for 25 s at 6 bar and 80 sccm, then manually transferred in 3 s to the 60 MHz benchtop NMR system for a 90° ^13^C pulse and signal acquisition. A 10 min delay between experiments allowed full thermal equilibration. The temperature at which each species reaches its maximum signal intensity—260 K for the bound **3b** and 272.5 K for free pyruvate—highlights the distinct exchange dynamics governing carbon hyperpolarization.

Precise timing of the bubbling process, enabled by the switch box, allows accurate estimation of the build-up time for ^13^C polarization. Figure 5b reports experiments at 280 K, 6 bar, and 80 sccm with bubbling times from 1 s to 180 s. Integration of the free pyruvate (left peak in the dashed subpanel in Figure 5b) and equatorially bound **3b** (right peak in Figure 5b) peaks yielded build-up times of 33.8 ± 3.8 s and 16.8 ± 3.3 s, respectively.

In addition, the effect of parahydrogen pressure at different temperatures was investigated for Ir–IMes, as shown in Figure 5c,d. The concentration of H_2_ in methanol-d_4_ solutions is expected to increase linearly with pressure within the explored temperature range [64]. Consistently, the signals corresponding to both free pyruvate (**4**) and the equatorially bound species (**3b**) increase with pressure, although the overall enhancement in signal intensity from 2 bar to 6 bar is only about 20%. However, the pressure dependence of the hyperpolarized signal can be significantly steeper for ^1^H species, underscoring the importance of accurately controlling this parameter.

The switch box (see Figure 1c,e,f and Figure 2) can be either operated manually as for the experiments at 1.4 T or automatically via the Topspin pulse program, as for the experiments at 9.4 T. This enables precisely controlled bubbling directly inside the NMR probe with a high magnetic field. Again, we tested the system using the Ir–SIPr and Ir–IMes catalysts. We first monitored the ^1^H hydride signal as a function of bubbling time to assess catalyst activation at 265 K, and once the catalyst activation was achieved, we investigated the polarization transfer from hydrides to carbon via the SEPP–INEPT pulse sequence [65,66,67].

When the spectra at 9.4 T are normalized to the most intense resonance in the 0–10 ppm range, two conclusions emerge. (i) Ir–SIPr exclusively exhibits **3b** hydrides (hydride protons corresponding to **3b** at around −27.2 ppm and −29.1 ppm), consistently with what we found in Figure 5a,b, while Ir–IMes shows both **3a** (hydride protons corresponding to **3a** at around −24.0 ppm and −14.9 ppm) and **3b** species (hydride protons corresponding to **3b** at around −27.2 ppm and −29.1 ppm), together with signals from **2** (hydride protons corresponding to **2** at around −15.5 ppm and −21.5 ppm), consistently with what has already been reported in the literature [54]. (ii) The integrated intensities of the **3b** hydrides indicate a ratio of approximately 6.7 between Ir–IMes and Ir–SIPr (Figure 6a,b), with an activation time for Ir–IMes of ~1 min at 265 K and 6 bar pressure (Figure 6c).

Overall, these results show that stronger σ-donation and steric bulk in Ir–SIPr stabilize hydride species but limit exchange dynamics, reducing pyruvate hyperpolarization efficiency. In contrast, the less hindered IMes ligand promotes faster exchange between **3a**, **3b**, and **2**, leading to higher polarization transfer at comparable conditions.

This mechanistic understanding also underscores the value of precisely controlled and automated experimental conditions, as subtle differences in ligand dynamics can only be probed reliably when gas flow, timing, and field exposure are accurately synchronized.

Building on these findings, the experimental setup allows polarization transfer experiments with a high magnetic field (9.4 T for us) to be conducted in a fully automated mode, offering the possibility of exploring pulse sequences designed to convert hydride-derived two-spin orders into heteronuclear magnetization. In particular, polarization can be transferred between the longitudinal two-spin order I_1z_I_2z_ (where I_1_ and I_2_ denote the nuclear hydrides’ spin) and the heteronuclear spin S_x_, the transverse spin operator associated with ^13^C in [1-^13^C]pyruvate) (Figure 6e).

One such experiment is the SEPP–INEPT sequence (Figure 6e), recently analyzed by Assaf et al. [66], which enables controlled transfer of hydride polarization to ^13^C. Without delving into technical details, the sequence converts hydride magnetization into ^13^C coherence. The 2S_y_F_z_ operator, antiphase magnetization on ^13^C spin, present before the 90° pulses on ^1^H and ^13^C is modulated by the *J*-coupling between the two hydrides during τ_1_ and by the *J*-coupling between one of the hydrides and ^13^C during τ_2_. Using the automated switch box, we implemented this sequence to acquire ^13^C spectra. For Ir–IMes, the polarization of the **3b** species was sufficiently high to detect the bound ^13^C resonance at 168.5 ppm (τ_1_ = 20 ms, τ_2_ = 60 ms), which can be compared to the one-pulse-detection ^13^C spectrum showing the free pyruvate signal at ~170 ppm (Figure 6e). In contrast, for Ir–SIPr, no detectable ^13^C signal was observed, consistent with the lower polarization level of the **3b** form evident in Figure 6a,b. By fixing τ_1_ at 20 ms and varying τ_2_ between 10 ms and 600 ms, we estimated the heteronuclear *J*-coupling introduced by Assaf et al. [65]. The signal was overall modulated according to the function Sin [2π *J*_SF_ τ_2_] × Exp[−2 τ_2_ R], where *R* represents the relaxation rate. The fitting of the experimental data employing the above function (blue line in Figure 6e) yielded a *J*-coupling between the hydride at −29.1 ppm and the ^13^C nucleus of bound (**3b**) pyruvate of approximately ~1.3 Hz. This value should be regarded as an estimate, whose accuracy can be improved by increasing the number of τ_2_ points used in the experiment.

These results confirm that efficient hydride–substrate exchange dynamics are a prerequisite for detectable heteronuclear polarization transfer, highlighting how the interplay between electronic donation and steric hindrance dictates both catalytic activity and polarization efficiency.

## 4. Materials and Methods

### 4.1. Sample Preparation

A solution containing 6 mM Ir-NHC catalyst, 20 mM sodium [1-^13^C]pyruvate (Merck SRL, CAS: 87976-71-4, from Merck Serono, Via Casilina 125, 00176 Roma, Italy), and 30 mM DMSO was prepared in methanol-d_4_. In the case of the 1.4 T experiments, the sample was bubbled with 50% parahydrogen at 6 bars for 20 min at ambient temperature for catalyst activation. In the case of the 9.4 T experiments, the sample was prepared in an identical way, but the activation was followed by repeated bubbling for 20 s for Ir–IMes. The Ir–SIPr and Ir–IMes were synthesized as described in the SI of reference [56].

### 4.2. μ-Metal, Flux-Gate Magnetometer, and Degaussing Wand

The three-layer mu-metal shield used in our setup ensures an internal magnetic field in the nanotesla range (resolution around 50 nT for our flux-gate magnetometer). Precisely controlled and accurate ultra-low (homogeneous) magnetic fields are essential for conducting SABRE-SHEATH experiments, which typically require sub-microtesla fields to efficiently hyperpolarize heteronuclei [68]. The desired magnetic field can be generated using a solenoid with sub-milliampere-level currents, as described above. In order to measure sub-microtesla magnetic fields, we used a flux-gate magnetometer described above (Bartington—MAG-03MC 1000; see Appendix A). A common issue to consider is the gradual magnetization of the mu-metal chamber itself over time, which can compromise field accuracy and homogeneity. In our experiments, even when using currents of a few amperes, we did not observe significant magnetization effects. Nevertheless, this risk can be mitigated by using a degaussing wand (see Appendix A). We also note that Chekmenev and collaborators have introduced a system in which the degaussing unit is integrated directly into the mu-metal shield, offering a more robust and automated solution [69].

### 4.3. NMR Tube–PTFE Connection Method

In all experiments we used standard Wilmad 5 mm outer diameter NMR tubes (5 mm O.D., 0.43 mm thickness; code: Z565229, vendor: https://www.sigmaaldrich.com/IT/it/product/aldrich/z565229, accessed on 1 November 2025).

These were connected to a 6 × 4 mm PTFE (https://ptfetubeshop.com/it/product/tubo-in-ptfe-4mm-x-6mm/, accessed on 1 November 2025) tube by gently heating the end of the PTFE tube with a heat gun to soften it. Once flexible, the upper part of the NMR tube was carefully inserted into the PTFE tube. Upon cooling, the PTFE forms a secure seal that was tested to withstand pressures of up to 8 bar during our experiments. The other end of the PTFE tube was connected to the Y push-in fitting, as schematically represented in Figure 1a.

### 4.4. SABRE-SHEATH Experiments

In SABRE-SHEATH experiments, the sample is manually transferred from the mu-metal shield to the benchtop NMR in approximately 3 s. The NMR tube and the gas distribution system function as a single integrated unit. The tube can be lifted simply by raising the connected 4 mm o.d. (i.d. 2.5 mm) polyurethane or nylon lines, which securely hold it in place through the push-in fitting connections. We verified that the PTFE seal between the NMR tube and the Y-push-in fitting (see Figure 1a) can withstand pressures of up to 8 bar, although all our experiments were conducted at 6 bar or below. Over the course of more than 200 experiments, we observed only one instance of the NMR tube detaching from the Y-fitting at 6 bar pressure. After the controlled bubbling step, the pressure can be released through the exhaust line, and the tube can be manually lifted and placed inside the benchtop NMR for signal detection. In the high-field experiments, the NMR tube—connected to the parahydrogen distribution system—is positioned in a high magnetic field and remains there throughout the experiment. The length of the gas lines connecting the tube to the distribution system can be adjusted as needed (approximately 4 m in our setup), depending on the spectrometer used, to ensure that the distribution system remains outside the 5 Gauss safety zone.

## 5. Conclusions

We have presented a cost-effective and versatile setup for performing parahydrogen-based hyperpolarization experiments at 50% parahydrogen enrichment under precisely controlled pressure, temperature, and flow conditions. The system integrates a custom-built switch box that can be operated either manually or automatically through TTL triggers directly synchronized with the NMR console, thus enabling fully automated experiments with high magnetic fields. The design is compatible with both benchtop and superconducting NMR instruments, extending the applicability of PHIP and SABRE techniques across a broad range of operating conditions.

The functionality of the setup has been demonstrated by investigating the hyperpolarization of sodium [1-^13^C]pyruvate in methanol-d_4_ using Ir–SIPr and Ir–IMes as catalysts. At 1.4 T, we performed variable-temperature experiments in manual mode using both Ir–SIPr and Ir–IMes, while precisely controlling the bubbling time and maintaining temperature stability within ±0.1 K. For Ir–IMes, we used the described setup to explore the dependence of the hyperpolarized signal not only on temperature but also on bubbling pressure, revealing the expected direct correlation between pressure and signal intensity [56].

The activation of the Ir–IMes catalyst was investigated at 9.4 T and 265 K, where we estimated the build-up time of the polarization via the hydride signals to be about 1 min. In addition, polarization transfer experiments from hydrides to ^13^C via the SEPP–INEPT sequence were demonstrated. We used this pulse sequence, following the work of Assaf et al., as a neat example of a non-trivial methodology to gain physical insights from hyperpolarization experiments [65,66,67].

Through the SEPP–INEPT sequence we have estimated the heteronuclear *J*-coupling between one of the hydrides and carbon 13 in the **3b** form to be in the 1 Hz range. Beyond demonstrating the feasibility of heteronuclear polarization transfer under SABRE conditions, the ability to run automated SEPP–INEPT experiments enables systematic optimization and quantitative analysis of spin transfer pathways, offering direct insight into the nature and dynamics of the active catalytic species.

Overall, the proposed system provides an accessible, reproducible, and scalable platform for PHIP and SABRE studies. By combining mechanical simplicity with electronic automation, it facilitates systematic investigations of catalyst performance and polarization dynamics—thereby contributing to the broader adoption and standardization of parahydrogen hyperpolarization methods in both research and teaching laboratories.

## Figures and Tables

**Figure 1 molecules-30-04299-f001:**
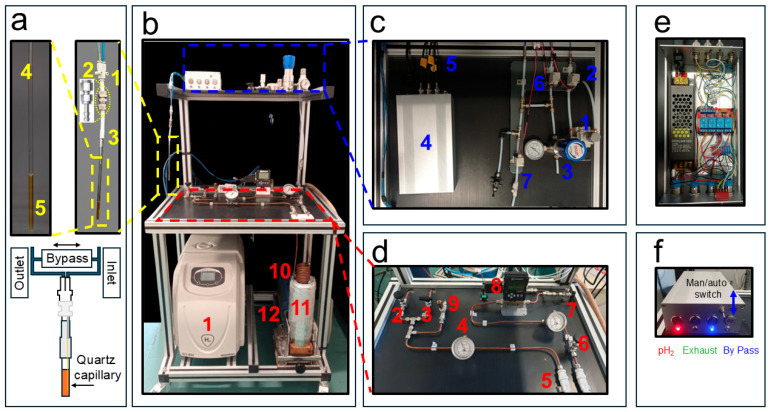
Description of the liquid-nitrogen parahydrogen generator and the gas distribution system. (**a**) The push-in Y fitting connecting the inlet and outlet gas lines (1), nickel-plated brass compression fittings module (2), PTFE tube (3), NMR tube (4), and quartz capillary for gas bubbling (5). (**b**) The H_2_ gas generator (1), the liquid nitrogen vessel (11) that can be lifted to immerse the 6 mm o.d. copper spiral (10) containing the Iron (III) catalyst for ortho- to parahydrogen conversion, and the diaphragm pump (12). (**c**) The distribution system with a pressure regulator (1), solenoid valves (2,6,7), a back-pressure regulator (3), switchbox (4), and BNC TTL cables to NMR console (5). (**d**) Expansion of the parahydrogen generator with two-way and single manual ball valve (2,3), pressure gauge (4), in-line microfilters (5), manual ball valve (6), pressure gauge (7), and mass-flow controller (8). In (**e**) Top and open view of the switchbox shown in Figure 1c(4). (**f**) Front view with push-button LED of the switch box in Figure 1c(4). Adapted from ref. [56] with permission from the Royal Society of Chemistry.

**Figure 2 molecules-30-04299-f002:**
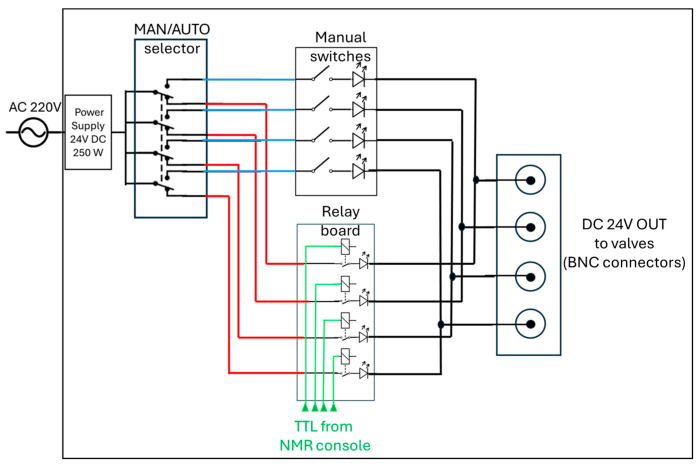
A schematic diagram of the control unit used to drive the solenoidal valves in the parahydrogen distribution section (Figure 1c(4)). The 24V DC used to switch four valves is routed by a manual switch (4PDT) either to a manual control set of buttons (manual switches) or to an automatic control system through an array of relays (relay board) controlled by the TTL signals coming from the NMR console (or from any TTL-compatible unit, e.g., Arduino). See also Figure 1e,f.

**Figure 3 molecules-30-04299-f003:**
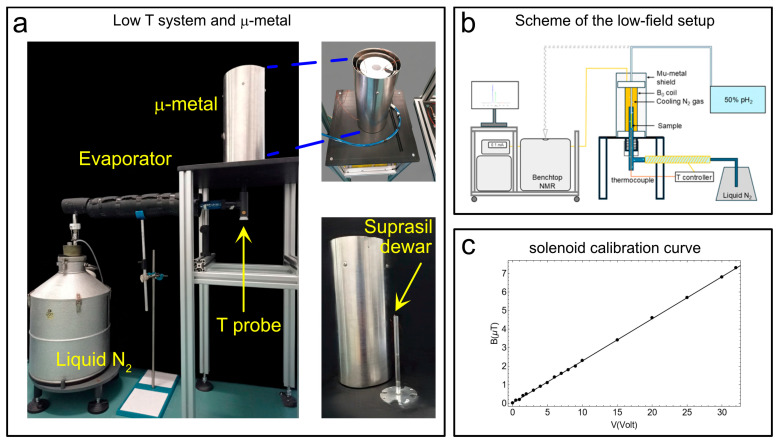
(**a**) The suprasil dewar, the μ-metal shield enclosing the solenoid coil to produce the B_0_ magnetic field in the 0 μT-to-10 μT range, the evaporator with temperature sensor (T-probe), and the liquid N_2_ dewar. (**b**) A simplified scheme of the experimental setup. (**c**) The calibration curve for the solenoid winding to produce the desired B_0_ in the μT range. Adapted from ref. [56] with permission from the Royal Society of Chemistry.

**Figure 4 molecules-30-04299-f004:**
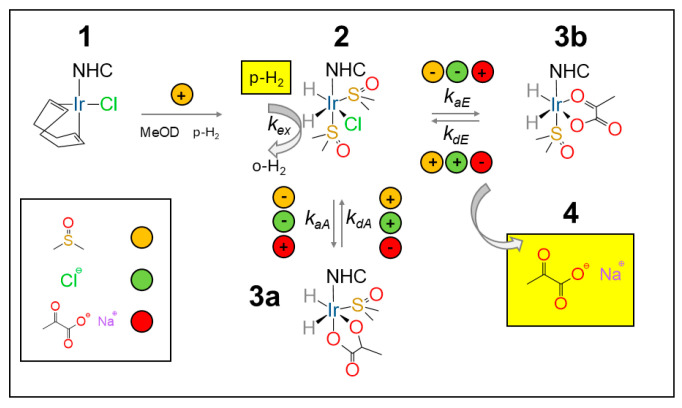
Pyruvate hyperpolarization by SABRE. The NHC precatalyst (**1**) is activated by H_2_ bubbling leading to the intermediate (**2**). The pyruvate molecule can either bind to the catalyst axially (**3a**) or equatorially (**3b**). Free pyruvate (**4**) is obtained in a reversible process via [Ir(H)_2_(κ^2^-pyruvate)(DMSO)(NHC)] (**3a**,**3b**), whereas hydrogen exchange is mediated by ([IrCl(H)_2_(DMSO)_2_(NHC)]) (**2**), as detailed by Duckett et al. in refs. [54,58]. Adapted from ref. [56] with permission from the Royal Society of Chemistry.

**Figure 5 molecules-30-04299-f005:**
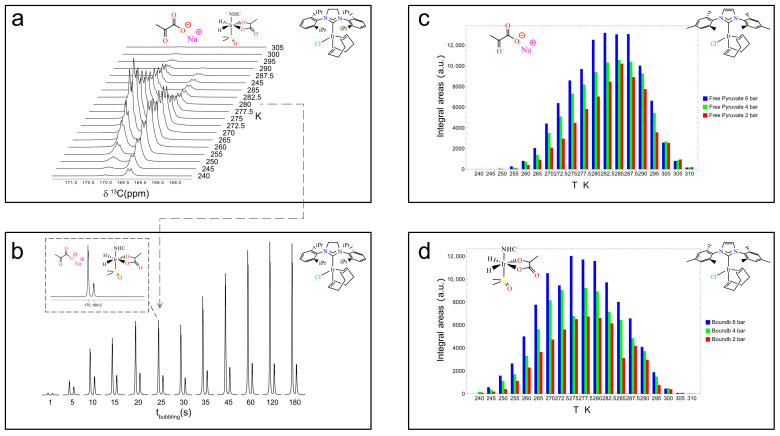
Experiments with Ir–SIPr catalyst. The sample includes 6 mM SIPr, 20 mM sodium [1-^13^C]pyruvate, and 30 mM DMSO for a total volume of 700 μL. The sample was bubbled for 25 s at 6 bars and 80 sccm at 0.35 μT and manually shuttled at 1.4 T for signal detection. The recovery time between experiments was set to 10 min. (**a**) Temperature profile experiments in the range of 240 K–305 K. (**b**) The bubbling time was varied from 1 s to 180 s in consecutive experiments to evaluate ^13^C hyperpolarization build-up kinetics. Ir–IMes temperature profiles in the 240 K-to-310 K range at parahydrogen bubbling pressures of 2 (red), 4 (green), and 6 (blue) bars in (**c**) for the free pyruvate form (**4**) and in (**d**) for the equatorially bound **3b** form. The data and figures in the subpanels were adapted from ref. [56] with permission from the Royal Society of Chemistry.

**Figure 6 molecules-30-04299-f006:**
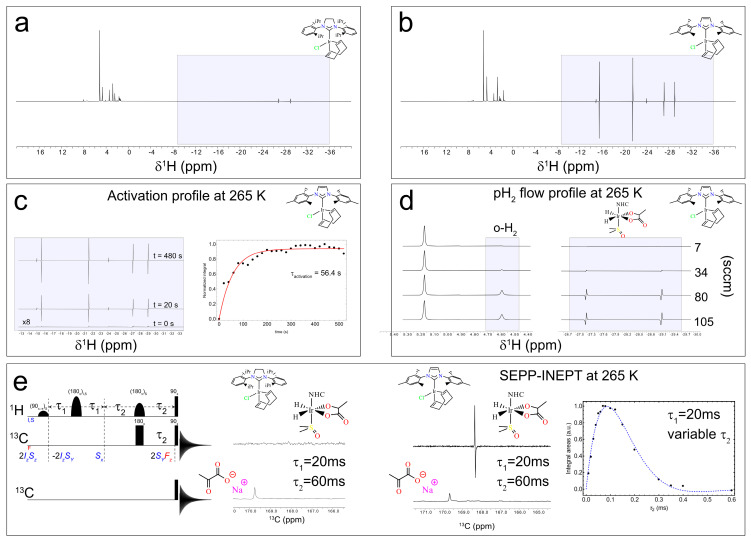
^1^H spectra acquired after 25 s of bubbling 50% parahydrogen with a 45° pulse for the solutions containing Ir–SIPr (**a**) and Ir–IMes (**b**). In the shaded box in (**a**), the signals correspond to species **3b**, with hydride resonances observed at approximately −27.2 ppm and −29.1 ppm. In the shaded box in (**b**), both **3a** (hydride protons at –24.0 ppm and −14.9 ppm) and **3b** species (hydride protons at −27.2 ppm and −29.1 ppm) are detected, together with signals from species **2** (hydride protons at −15.5 ppm and −21.5 ppm). (**c**) Activation profile of Ir–IMes obtained by progressively bubbling H_2_ into the solution and monitoring the hydride ^1^H signal. The data points represent the integrals of one of the two hydride **3b** resonances. (**d**) ^1^H spectra of the dissolved H_2_ region (≈4.59 ppm) and the hydride region of **3b** recorded at increasing flow rates from 7 sccm to 105 sccm. (**e**) SEPP–INEPT pulse sequence with the corresponding quantum operators (*I*,*S* indicates hydrides and *F* indicates ^13^C) at different steps and ^13^C pulse-acquisition sequence, along with the ^13^C spectra at 265 K for Ir–SIPr and Ir–IMes using τ_1_ = 20 ms and τ_2_ = 60 ms. By varying τ_2_, the integral of the **3b** peak is plotted, allowing estimation of the J-coupling between the hydride at −29.2 ppm and the ^13^C at 168.5 ppm.

## Data Availability

NMR data for this article are available at Zenodo at https://doi.org/10.5281/zenodo.17502107, accessed on 1 November 2025.

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
