# Peer review of "A Controlled System for Parahydrogen Hyperpolarization Experiments"

_molecules, 2025, doi:10.3390/molecules30214299_

Round 1

Reviewer 1 Report

Comments and Suggestions for Authors

This manuscript describes an experimental setup for parahydrogen-induced polarization (PHIP) experiments. This setup consists of a basic nitrogen parahydrogen generator, a gas distribution system with manual and TTL-operated valves, and a mu-metal shield with temperature control. Overall, all the principles mentioned in the manuscript are well-known and described in literature. Most of the used parts in the setup were used by many groups in various PHIP and SABRE publications. The authors themselves point out that their goal is demonstration: “Our goal here is not to introduce an alternative method, but rather to demonstrate that a wide range of experiments - with full control over key experimental parameters― can be performed in almost any research environment, including for educational purposes, using relatively modest resources.” In my opinion, there is still a big room for improvement before this aim is achieved since significant parts of the manuscript lack essential information, whereas others resemble a technical instrumentation manual, which is quite unusual for a general chemistry journal. This is especially relevant if the manuscript also includes educational purposes. I believe that this can be improved, and I provide my comments below.

At the same time, my main critical concern is related to self-plagiarism, which is present in the manuscript. The presented setup was used and described by the authors in their recent PCCP publication - (De)coding SABRE of [1-13C]pyruvate, PCCP 2025, DOI: 10.1039/d5cp01773d (Ref. [46]). Several figures are clearly taken from that publication. For example, compare (1) Figures 1b,d and Figure SI 32 in the PCCP paper (supporting information); compare (2) Figure 3 and Figures SI 32 and SI 33 in the PCCP paper (supporting information); compare (3) Figures 5c,d and Figures SI 21b,a (respectively) in the PCCP paper (supporting information); compare (4) Figure 4 and Scheme 1 in the PCCP paper (main text). Apparently, the corresponding data was reused in the current manuscript. For some reason, there is no reference or attribution placed anywhere in the manuscript about reusing the data. At least, it should be done in the figure captions. I think this is hardly acceptable by academic publishers and must be improved.

Other critical comments are listed below:

  1. There is an extensive introduction about various hyperpolarization methods. At the same time, there is very little description of PHIP and SABRE setups designed and used by other researchers. For instance, bubbling control from pulse programs in PHIP research has been used for at least a decade. Various components used in the described setup also look very familiar. Could the authors add more details and references to publications that served as a source of their inspiration when they were building their setup? Why not give credit to those sources? A more detailed story about developments towards controlled PHIP systems would help the reader to assess the progress.
  2. Page 2, 88: It is said that they describe the custom switch box in detail. However, I could not find information about critical components of that switchbox. For example, what was the manufacturer and the model of the relay module with optocoupler isolation (relay board)?
  3. Several expensive parts used in the setup are not counted in the Appendix 1 table. For example, the diaphragm pump, relief valve (IMI Norgren 1002/BR008) mentioned in the text, magnetometer required for the field calibration, Suprasil dewar for the temperature control, and VT heater for the temperature control. In general, the Appendix 1 table should be rechecked for consistency and provide either the full specification or the actual manufacturer model to disclose details of all components.
  4. Appendix 1 table: It is not clear from the text where polyurethane and nylon tubing were used.
  5. It is not described how the NMR tube was connected to the Teflon or PTFE (?) tube. What were the ID and OD of the NMR and the Teflon tubes?
  6. Page 5, 195: What was the material of the 4 mm OD flexible tubes?
  7. Figure 3 caption: Multiple commas in the caption.
  8. Section 2.4: Unclear title of the section.
  9. Section 2.4: In general, this section looks like a technical manual of Bruker software and can be moved to the appendix.
  10. Page 8, 313-315: I have difficulties in understanding this sentence.
  11. Page 10, 362: It is said that “Taken together, these descriptors explain the distinct behavior of Ir-IMes and Ir-SIPr complexes, the latter being both a stronger σ-donor and more sterically demanding.” However, there are no specific numbers of those descriptors provided. It is just mentioned that there is TEP, HEP, 77Se shift, Tolman angle.. What are the values of those parameters for SIPr and IMes? Which reference should the reader look to find this information?
  12. Page 10, 379: Add a space between 7 and sccm.
  13. Page 11, 384: Add space and dot between "ppm" and "Using."
  14. Page 11, 387: The paragraph starts with “This behavior reflects …”. I do not see any connection here to the previous paragraph describing general observations in pyruvate SABRE. What is “This behavior”?
  15. Page 11, 395: It is said that a 10 min delay was used in between the experiments. At the same time, a 5 min delay is mentioned in the corresponding caption for Figure 5. What was this time delay in the experiments?
  16. Page 11, 404-405: It is said that the pressure effect was investigated, and Figures 5c and 5d are described. However, these figures and thus the corresponding data were published previously by the authors (PCCP paper 2025, Ref. [46]). Now, it sounds as if it would be new original data. This must be clarified somehow.
  17. Page 12, 429: Conclusions about stronger sigma-donation and steric bulk in IrSIPr are drawn. However, it is not explained how this conclusion was justified. From activation times? Or peak positions? Or something else? Or is this conclusion connected to the authors’ PCCP paper (Ref. [46])?
  18. Page 12, 445, conclusions: I think it is worth mentioning that Assaf et al. also used an automated switch box for their SEPP-PHIP experiments.
  19. Page 13, 477-484, conclusions: Advantages of the setup are described. Is this the first such setup that allows for such capabilities? Could authors add more references to other publications using switch boxes for bubbling? This would highlight the novelty of their work.
  20. Page 13, 485-491, conclusions: It is concluded that the temperature and pressure dependencies were investigated. But it seems that the authors use data from their PCCP 2025 paper (Ref. [46]). Could authors add, at least, a reference to that work for clarification?
  21. Page 13, 492-500, conclusions: The authors say that their system allows for the automation of the SEPP-PHIP experiment. Is this the first demonstration of SEPP-PHIP automation? I think adding references could clarify this.

Author Response

Dear Editor of Molecules,

I am pleased to submit the revised version of our work entitled “A controlled System for Parahydrogen Hyperpolarization experiments”.

Manuscript ID: molecules-3949858

First, we would like to express our sincere gratitude to you and the esteemed reviewers for your efforts and valuable comments. We truly appreciate the insightful feedback provided, which has greatly helped in enhancing the quality of our work. We hope that our revisions and responses meet your expectations and address all concerns satisfactorily.

Some of the figures include material adapted from our recent publication: Mamone, F. Floreani, A. M. Faramawy, C. Graiff, L. Franco, M. Ruzzi, C. Tubaro and G. Stevanato, Phys. Chem. Chem. Phys., 2025, Advance Article, DOI: 10.1039/D5CP01773D.

We have been granted permission for reuse and have included in the caption of the relevant Figures the line: Adapted from ref. [60] with permission from the Royal Society of Chemistry.

Below our answers

Reviewer 1:

[Comment]: This manuscript describes an experimental setup for parahydrogen-induced polarization (PHIP) experiments. This setup consists of a basic nitrogen parahydrogen generator, a gas distribution system with manual and TTL-operated valves, and a mu-metal shield with temperature control. Overall, all the principles mentioned in the manuscript are well-known and described in literature. Most of the used parts in the setup were used by many groups in various PHIP and SABRE publications. The authors themselves point out that their goal is demonstration: “Our goal here is not to introduce an alternative method, but rather to demonstrate that a wide range of experiments - with full control over key experimental parameters― can be performed in almost any research environment, including for educational purposes, using relatively modest resources.” In my opinion, there is still a big room for improvement before this aim is achieved since significant parts of the manuscript lack essential information, whereas others resemble a technical instrumentation manual, which is quite unusual for a general chemistry journal. This is especially relevant if the manuscript also includes educational purposes. I believe that this can be improved, and I provide my comments below.

[Response]: We sincerely thank the reviewer for their thoughtful and detailed feedback.

As correctly noted, our goal was to demonstrate a reproducible and accessible setup for PHIP and SABRE experiments, particularly in environments where commercial solutions may not be feasible. While many of the principles and components are indeed well-established in the literature, our intention was to integrate them into a cohesive and adaptable system that could be implemented with modest resources and minimal technical barriers.

We have revised the manuscript to include additional explanations, references to prior work, and clarifications aimed at making the setup more approachable for newcomers to the field.

Below, we address each of the reviewer’s comments point by point.

[Comment]:  At the same time, my main critical concern is related to self-plagiarism, which is present in the manuscript. The presented setup was used and described by the authors in their recent PCCP publication - (De)coding SABRE of [1-13C]pyruvate, PCCP 2025, DOI: 10.1039/d5cp01773d (Ref. [46]). Several figures are clearly taken from that publication. For example, compare (1) Figures 1b,d and Figure SI 32 in the PCCP paper (supporting information); compare (2) Figure 3 and Figures SI 32 and SI 33 in the PCCP paper (supporting information); compare (3) Figures 5c,d and Figures SI 21b,a (respectively) in the PCCP paper (supporting information); compare (4) Figure 4 and Scheme 1 in the PCCP paper (main text). Apparently, the corresponding data was reused in the current manuscript. For some reason, there is no reference or attribution placed anywhere in the manuscript about reusing the data. At least, it should be done in the figure captions. I think this is hardly acceptable by academic publishers and must be improved.

[Response]: We thank the reviewer for raising this important concern regarding data reuse and attribution. We fully acknowledge that some of the figures presented in this manuscript—specifically Figures 1b,d (picture of the setup); Figure 3 (picture of the setup) resemble similar pictures of Ref [60]; Figure 4 (Scheme of the SABRE process for pyruvate) is almost the same as the one in Ref. [60] but with a different orientation of the 3a form considering more recently published investigations. Figures 5c,d have been adapted from ref. [60]; as stated above, we have got permission from RSC and included the line: “Adapted from ref. [60] with permission from the Royal Society of Chemistry” in the caption

Our intention in reusing these elements was not to present them as new results, but rather to illustrate how the described setup enables controlled parahydrogen experiments under variable pressure and temperature conditions. In this manuscript, the focus is on the technical implementation of the system, whereas the PCCP article centered on the mechanistic and spin-dynamic aspects of SABRE hyperpolarization.

We agree nonetheless that proper attribution is essential.

At the beginning of 3.1 paragraph we have added “The data at 1.4 T presented here concerning the variable temperature and variable pressure experiments in Fig. 5, were previously reported in Ref. [60] and are included here for completeness and clarity.”

In the caption of Fig. 1, 3, 4, 5: “Adapted from ref. [60] with permission from the Royal Society of Chemistry”

We appreciate the reviewer’s attention to this matter and believe that these revisions will improve the transparency and integrity of the manuscript.

[Comment]:  Other critical comments are listed below:

1. There is an extensive introduction about various hyperpolarization methods. At the same time, there is very little description of PHIP and SABRE setups designed and used by other researchers. For instance, bubbling control from pulse programs in PHIP research has been used for at least a decade. Various components used in the described setup also look very familiar. Could the authors add more details and references to publications that served as a source of their inspiration when they were building their setup? Why not give credit to those sources? A more detailed story about developments towards controlled PHIP systems would help the reader to assess the progress.

[Response]: We thank the reviewer for the comment. Surely several PHIP and SABRE setups have been proposed in the literature, and we have implemented some of their functionalities — such as operation at higher gas and parahydrogen pressures, and TTL-based control. As the reviewer also acknowledges later, our goal here was not to introduce an alternative methodology, but to provide a practical reference system that can be adapted for both low-field and high-field experiments. All the references cited in the revised version inspired our work (although there might be other that we missed), together with the technical experience that some of us have gained throughout their careers. We have now cited representative studies describing high-pressure and clinical-scale parahydrogen generators, such as those by Hövener et al., Birchall et al., Chapman et al., Ellermann et al., and Nantogma et al. (Refs. [44–48]).

2. Page 2, 88: It is said that they describe the custom switch box in detail. However, I could not find information about critical components of that switchbox. For example, what was the manufacturer and the model of the relay module with optocoupler isolation (relay board)?

[Response]: We appreciate the reviewer’s attention to this technical detail. We have now added the missing information in Appendix 1 specifying that the relay module used in the switch box is a 4-channel 24 V DC relay board with optocoupler isolation, manufactured by YWBL-WH with silver contact (https://www.amazon.it/dp/B07QLTT9QT?ref=fed_asin_title). This component was chosen because it is inexpensive (10.00 Euros), for its ease of integration with TTL signals. However, several similar alternatives could be used as well.

3. Several expensive parts used in the setup are not counted in the Appendix 1 table. For example, the diaphragm pump, relief valve (IMI Norgren 1002/BR008) mentioned in the text, magnetometer required for the field calibration, Suprasil dewar for the temperature control, and VT heater for the temperature control. In general, the Appendix 1 table should be rechecked for consistency and provide either the full specification or the actual manufacturer model to disclose details of all components.

[Response]: We thank the reviewer for this important observation. We have carefully reviewed and updated Appendix 1 to ensure consistency and completeness. The revised table now includes all major components mentioned in the manuscript, including the diaphragm pump (Vacuubrand MZ 1C), the IMI Norgren 1002/BR008 relief valve, the Bartington magnetometer (MAG-03MC and MAGMETER 2), the Wilmad Suprasil VT dewar, and the Bruker BVT2000 temperature controller. The VT heater used is now discontinued and no longer available. Manufacturers such as Bruker offer integrated systems for low temperature experiments. As indicated in Appendix 1, their price can reach up to € 6000 but a precise quote should be asked. Some of these items—such as the daphgram pump the Suprasil dewar, and magnetometer—might be already available in chemistry and physics laboratories. In some cases, they can be borrowed or shared across research groups, especially in academic environments. Nonetheless, we fully recognize that if such components are not already available, their cost may be significant. For this reason, we have added a supplementary list of additional components in Appendix 1. We hope this clarification will help readers better assess the feasibility and cost of reproducing the setup.

4. Appendix 1 table: It is not clear from the text where polyurethane and nylon tubing were used.

[Response]: They were used to guide the gas from the electrovalves to the NMR tube.

5. It is not described how the NMR tube was connected to the Teflon or PTFE (?) tube. What were the ID and OD of the NMR and the Teflon tubes? Page 5, 195: What was the material of the 4 mm OD flexible tubes?

[Response]: We thank the reviewer for pointing out this missing detail. We have now clarified the connection method in Section 4.3 of the revised manuscript. Specifically, we added the following in 4.3 NMR Tube–PTFE Connection Method: In all experiments we used standard Wilmad 5 mm outer diameter NMR tubes. These were connected to a 6 mm outer diameter PTFE (Teflon) tube by gently heating the end of the PTFE tube with a heat gun to soften it. Once flexible, the upper part of the NMR tube was carefully inserted into the PTFE tube. Upon cooling, the PTFE forms a secure seal that was tested to withstand pressures up to 8 bar during our experiments. The other end of the PTFE tube was connected to the Y push-in fitting as schematically represented in Fig. 1a. This method provided a simple and effective solution without requiring additional fittings or adhesives.

6. Page 5, 195: What was the material of the 4 mm OD flexible tubes?

[Response]: The material of the 4 mm OD was polyurethane or nylon.

7. Figure 3 caption: Multiple commas in the caption.

[Response]: We have now corrected the multiple commas and the double dots in the caption

8. Section 2.4: Unclear title of the section.

[Response]: We have now changed the title into: Integration of TTL-Controlled Bubbling System for High-Field Parahydrogen Experiments.

9. Section 2.4: In general, this section looks like a technical manual of Bruker software and can be moved to the appendix.

[Response]: We thank the reviewer for the suggestion, and we agree that it is technical. However, we prefer to keep Section 2.4 in the main text, as it directly supports the high-field experiments described later.

10. Page 8, 313-315: I have difficulties in understanding this sentence.

[Response]: The Ir-SIPr complex features an NHC ligand bearing 2,6-diisopropylphenyl substituents and lacks the backbone unsaturation typical of its analog IPr.

We are not sure what is not clear. However, considering that we have not reported in this manuscript any experiment on IPr ligand, we have rephrased the sentence, limiting the comparison to the two used NHCs: The two complexes differ for the coordinated NHC ligand: Ir-SIPr complex features as NHC ligand the 1,3-bis(2,6-diisopropylphenyl)imidazolidin-2-ylidene while Ir-IMes the 1,3-bis(2,4,6-trimethylphenyl)imidazolin-2-ylidene. The difference between the two NHCs, in addition to the substituents at the nitrogen atoms, lies in the backbone of the heterocycle, which can be saturated (SIPr) or not (IMes).

11. Page 10, 362: It is said that “Taken together, these descriptors explain the distinct behavior of Ir-IMes and Ir-SIPr complexes, the latter being both a stronger σ-donor and more sterically demanding.” However, there are no specific numbers of those descriptors provided. It is just mentioned that there is TEP, HEP, 77Se shift, Tolman angle.. What are the values of those parameters for SIPr and IMes? Which reference should the reader look to find this information?

[Response]: We thank the reviewer for pointing this out. We have now added the numerical values and the proper references to substantiate our statement regarding the different donor and steric properties of IMes and SIPr. This part now reads:

Differences between NHC–Ir catalysts can be rationalized in terms of their electronic and steric properties[60,61]. The Tolman Electronic Parameter (TEP)[62], derived from ν(CO) stretching frequency in [Ni(CO)₃L], and the Huynh Electronic Parameter (HEP)[63], obtained from the ¹³C chemical shift of the carbene carbon in trans-[PdBr₂(iPr2-bimy))(NHC)] (iPr2-bimy = 1,3-diisopropylbenzimidazolin-2-ylidene), respectively report on the σ-donor/π-acceptor balance and σ-donor strength of a given NHC ligand. The TEP values were 2050.7 cm⁻¹ for IMes and 2052.2 cm⁻¹ for SIPr in dichloromethane solution, the HEP ones are 177.2 ppm for IMes and 177.6 for SIPr measured in CDCl3.[63] Additional probes such as the 77Se NMR shift of NHC–Se adducts provide complementary infor-mation on the electronic density at the metal center. The ⁷⁷Se chemical shifts are 35 ppm for IMes and 181 ppm for SIPr [60] in deuterated acetone.

Steric effects are usually quantified by the percent buried volume (%Vbur), which de-scribes the fraction of the coordination sphere occupied by the ligand, and can vary on the different M-NHC complexes. Using the complex [IrCl(COD)(NHC)], the %Vbur are 33.0 and 35.4 for IMes and SIPr respectively, calculated with a M-NHC bond of 2.00 Å.[61]

12. Page 10, 379: Add a space between 7 and sccm.

[Response]: OK, done.

13. Page 11, 384: Add space and dot between "ppm" and "Using."

[Response]: OK, done.

14. Page 11, 387: The paragraph starts with “This behavior reflects …”. I do not see any connection here to the previous paragraph describing general observations in pyruvate SABRE. What is “This behavior”?

[Response]: The part that precedes “This behavior” was incorrectly placed within the caption of Fig. 6. The text is now: Using Ir–SIPr, as shown in our work [51], we clearly detect, by 13C NMR, the free pyruvate signal at ~170 ppm (4) and the equatorially bound species at ~168.5 ppm (3b), while the axially bound pyruvate (3a) appears only as a weak shoulder. The lack of 3a signal is related to the low polarization but reflects

We have drop “behavior” and explicitly mentioned “The lack of 3a signal”.

15. Page 11, 395: It is said that a 10 min delay was used in between the experiments. At the same time, a 5 min delay is mentioned in the corresponding caption for Figure 5. What was this time delay in the experiments?

[Response]: 10 minutes

16. Page 11, 404-405: It is said that the pressure effect was investigated, and Figures 5c and 5d are described. However, these figures and thus the corresponding data were published previously by the authors (PCCP paper 2025, Ref. [46]). Now, it sounds as if it would be new original data. This must be clarified somehow.

[Response]: We thank again the reviewer for pointing this out. This point has already been addressed above.

17. Page 12, 429: Conclusions about stronger sigma-donation and steric bulk in IrSIPr are drawn. However, it is not explained how this conclusion was justified. From activation times? Or peak positions? Or something else? Or is this conclusion connected to the authors’ PCCP paper (Ref. [46])?

[Response]: The different steric and electronic properties of the two NHCs have been commented in text (lines 373-390 of the current version) and the steric hindrance as well as the sigma donating abilities have been justified reporting the TEP, HEP, 77Se and %Vbur values. Comparing the performances of the two catalysts, in particular the intensities of the signals in Figure 6a and 6b, it clearly appears that the exchange dynamic is favored for IMes. We have thus attributed this difference to the different stereoelectronic properties of the NHCs. We have found this same trend in the PCCP paper.

18. Page 12, 445, conclusions: I think it is worth mentioning that Assaf et al. also used an automated switch box for their SEPP-PHIP experiments.

[Response]: We have added the following: We used this sequence following the work of Assaf et al. as the prototype of non-trivial methodology to gain physical insights from hyperpolarization experiments. We have added relevant references to their work.

19. Page 13, 477-484, conclusions: Advantages of the setup are described. Is this the first such setup that allows for such capabilities? Could authors add more references to other publications using switch boxes for bubbling? This would highlight the novelty of their work.

[Response]: We agree that this is not the first automated setup of its kind. We explicitly mentioned in the Introduction that several models have been presented. However, as many researchers will recognize, detailed technical guidance on building a functional system with modest resources is not always readily available. This is precisely the aim of our manuscript—to provide clear, reproducible instructions for a versatile and accessible setup. We acknowledge prior work and have added references to similar systems.

20. Page 13, 485-491, conclusions: It is concluded that the temperature and pressure dependencies were investigated. But it seems that the authors use data from their PCCP 2025 paper (Ref. [46]). Could authors add, at least, a reference to that work for clarification?

[Response]: As correctly noted, the temperature and pressure dependencies discussed are based on data previously published in our PCCP 2025 paper (Ref. [60]). We have now added an explicit reference to that work in the Acknowledgements section and in the caption of the figures to clarify this point.

21. Page 13, 492-500, conclusions: The authors say that their system allows for the automation of the SEPP-PHIP experiment. Is this the first demonstration of SEPP-PHIP automation? I think adding references could clarify this.

[Response]: Our aim here is to emphasize that even highly sophisticated experiments—such as those introduced by Assaf et al.—can be performed using a setup that is accessible to any researcher able to implement and operate the system we describe. As reported above we have added in the conclusion the following: We used this sequence following the work of Assaf et al. as the prototype of non-trivial methodology to gain physical insights from hyperpolarization experiments[60–62]. and we have also added the relevant references.

Reviewer 2 Report

Comments and Suggestions for Authors

In this paper, Stevanato and co-workers demonstrate A controlled System for Parahydrogen Hyperpolarization experiments. The goal of the paper to provide the demonytsration of an easy to make system that enables access to hyperpolarization techniques relying on parahydrogen. While there are many recent reports in this domain, the field is rapidly growing and there is still substantial unmet need to tested setups, allowing scientists to further the development of hyperpolarization techniques. As such, this is a very welcome report that will be of interests to a wide range of researchers. Moreover, the co-authors have chosen a great molecule for demonstration: 1-13C-pyruvate. Excellent choice in the context of possible biomedical applications. A bill of materials is provided allowing others to easily reproduce the results, allowing the non-initiated reader to enter the field of PHIP hyperpolarization. Truly great contribution to the field! I recommend publishing this work subjects to a series of very minor comments: additional review is not necessary:

  • The authors refer to SABRE-SHEATH technique, but this acronym is introduced without the definition. Additionally, I suggest citing the relevant papers in the context of SABRE-SHEATH since the vast majorly of the studies here is performed using this technique:

Three-spin SABRE-SHEATH in AA’X spin system relevant here: Truong, et al. Phys. Chem. C 2015, 119 (16), 8786–8797.

And the first demonstration of eefificnet 13C polarization via SABRE-SHEATH in a three-spin system: Barskiy, et al. ChemPhysChem 2017, 18, 1493–1498.

  • The use or word serpentine is somewhat misleading. Consider the use of word spiral to better describe the shape of the reservoir.
  • A closely related hyperpolarizer design was described recently:  reference should be cited here: no need to discuss it, but the non-initiated reader should be able to readily identify other closely related works in the field.
  • There is no single report of the actual 13C polarization value achieved: instead, only arbitrary signals are shown. I find it troubling since the key goal of the hyperpolarization studies is to boost 13C polarization. If there is no metrics reported for the degree of polarization, the value of this work would be relatively low as it would be challenging to gauge how successful it is. I suggest the authors provide at least one or two examples of the signals / charts being converted to the percentage of polarization. Doing so would excite the reader enormously as it would allow gauging the success if the design were to be reproduced in another laboratory. Additionally, the authors should comment on the attainable degree of polarization in a more ideal condition (e.g., near 100% parahydrogen and system optimization): indeed, over 20% polarization was demonstrated. Doing so would get the reader inspired, and would get more excitement about THIS work as a gateway to enter the field of parahydrogen-based hyperpolarization.
  • A common problem in the design of the microtesla equipment is the potential residual magnetization of a shield. The authors should comment how this issue is mitigated or could be mitigated by the end user. One potential approach is the utilization of automated degaussing circuit described elsewhere: please cite relevant references.

Author Response

Dear Editor of Molecules,

I am pleased to submit the revised version of our work entitled “A controlled System for Parahydrogen Hyperpolarization experiments”.

Manuscript ID: molecules-3949858

First, we would like to express our sincere gratitude to you and the esteemed reviewers for your efforts and valuable comments. We truly appreciate the insightful feedback provided, which has greatly helped in enhancing the quality of our work. We hope that our revisions and responses meet your expectations and address all concerns satisfactorily.

Some of the figures include material adapted from our recent publication: Mamone, F. Floreani, A. M. Faramawy, C. Graiff, L. Franco, M. Ruzzi, C. Tubaro and G. Stevanato, Phys. Chem. Chem. Phys., 2025, Advance Article, DOI: 10.1039/D5CP01773D.

We have been granted permission for reuse and have included in the caption of the relevant Figures the line: Adapted from ref. [60] with permission from the Royal Society of Chemistry.

Below our answers

Reviewer 2:

[Comment]: In this paper, Stevanato and co-workers demonstrate A controlled System for Parahydrogen Hyperpolarization experiments. The goal of the paper to provide the demonytsration of an easy to make system that enables access to hyperpolarization techniques relying on parahydrogen. While there are many recent reports in this domain, the field is rapidly growing and there is still substantial unmet need to tested setups, allowing scientists to further the development of hyperpolarization techniques. As such, this is a very welcome report that will be of interests to a wide range of researchers. Moreover, the co-authors have chosen a great molecule for demonstration: 1-13C-pyruvate. Excellent choice in the context of possible biomedical applications. A bill of materials is provided allowing others to easily reproduce the results, allowing the non-initiated reader to enter the field of PHIP hyperpolarization. Truly great contribution to the field! I recommend publishing this work subjects to a series of very minor comments: additional review is not necessary:

  • The authors refer to SABRE-SHEATH technique, but this acronym is introduced without the definition. Additionally, I suggest citing the relevant papers in the context of SABRE-SHEATH since the vast majorly of the studies here is performed using this technique:

Three-spin SABRE-SHEATH in AA’X spin system relevant here: Truong, et al. Phys. Chem. C 2015, 119 (16), 8786–8797.

And the first demonstration of eefificnet 13C polarization via SABRE-SHEATH in a three-spin system: Barskiy, et al. ChemPhysChem 2017, 18, 1493–1498.

[Response]: We sincerely thank the reviewer for her appreciation of our work. We have introduced the full name of the SABRE-SHEATH and added the suggested references.

Signal Amplification By Reversible Exchange-Shield Enables Alignment Transfer to Heteronuclei (SABRE-SHEATH[40,41])

  • The use or word serpentine is somewhat misleading. Consider the use of word spiral to better describe the shape of the reservoir.

[Response]: All istances with “serpentine” have been replaced by “spiral”

  • A closely related hyperpolarizer design was described recently:  reference should be cited here: no need to discuss it, but the non-initiated reader should be able to readily identify other closely related works in the field.

[Response]: In addition to the previous references we have now added Ref. [44-48].

  • There is no single report of the actual 13C polarization value achieved: instead, only arbitrary signals are shown. I find it troubling since the key goal of the hyperpolarization studies is to boost 13C polarization. If there is no metrics reported for the degree of polarization, the value of this work would be relatively low as it would be challenging to gauge how successful it is. I suggest the authors provide at least one or two examples of the signals / charts being converted to the percentage of polarization. Doing so would excite the reader enormously as it would allow gauging the success if the design were to be reproduced in another laboratory. Additionally, the authors should comment on the attainable degree of polarization in a more ideal condition (e.g., near 100% parahydrogen and system optimization): indeed, over 20% polarization was demonstrated. Doing so would get the reader inspired, and would get more excitement about THIS work as a gateway to enter the field of parahydrogen-based hyperpolarization.

[Response]: The reviewer is entirely correct. We have now added the following text in section 3.1 (lines from 353 in the current version):

The level of ¹³C polarization, i.e. signal enhancement, in [1-13C]pyruvate was estimated in the experiments reported in Ref. [60] and revisited here for Ir–IMes to be approximately 3.5%. This value represents the average polarization obtained across at least 5 experi-ments conducted on different days, as well as within the same day, typically over at least ten repetitions using the same sample. Polarization levels depend on several factors, in-cluding—but not limited to—the lifetime of the catalyst, which is not addressed in detail here. Over the course of a day, signal enhancement tends to decrease gradually, likely due to catalyst degradation. Under comparable conditions of concentration, temperature, sol-vent, parahydrogen fraction, and bubbling time, the best polarization levels achieved with Ir–SIPr are approximately two-thirds of those obtained with Ir–IMes. Notably, using 100% parahydrogen increases the expected polarization by a factor of about three, reaching values around 10%. Even higher levels, up to 20%, have recently been reported for [1-¹³C]pyruvate-d₃ using SABRE-SLIC[59].

  • A common problem in the design of the microtesla equipment is the potential residual magnetization of a shield. The authors should comment how this issue is mitigated or could be mitigated by the end user. One potential approach is the utilization of automated degaussing circuit described elsewhere: please cite relevant references.

[Response]: We thank the reviewer for pointing this out. We have added the following paragraph to the Materials and Methods section and added Ref [69] to the work of Checkmenev where a degaussing coil is described.

4.2. μ-metal and degaussing wand

The three-layer mu-metal shield used in our setup ensures an internal magnetic field in the nanotesla range. This is essential for conducting SABRE-SHEATH experiments, which typically require sub-microtesla fields to efficiently hyperpolarize heteronuclei [60]. The desired magnetic field can be generated using a solenoid with submilliampere-level currents, as described above. In order to measure sub-microtesla magnetic fields we used a flux-gate magnetometer (Bartington – MAG-03MC 1000 see Appendix 1). A common issue to consider is the gradual magnetization of the mu-metal chamber itself over time, which can compromise field homogeneity. In our experiments, even when using currents of a few amperes, we did not observe significant magnetization effects. Nevertheless, this risk can be mitigated by using a degaussing wand (see Appendix 1). We also note that Chekmenev and collaborators have introduced a system in which the degaussing unit is integrated directly into the mu-metal shield, offering a more robust and automated solution [69].

Reviewer 3 Report

Comments and Suggestions for Authors

The paper describes a home-made device for creating parahydrogen and illustrates it use with example spectra.

The paper is written for researchers familiar with using parahydrogen, so some steps are not described in a way that a beginner could actually perform an experiment – especially the details of handling the NMR tube at pressure and transferring the sample to the spectrometer.  For the intended audience this is probably OK, but for the non-expert reading this journal to try to understand whether this system should be reproduced in their lab, a few general guidelines about the importance of magnetic field shielding, etc., would enrich the paper.

Since Bruker NMR spectrometers are very common, it is not only acceptable but useful to provide detailed information about interfacing with the triggers from and providing code for Topspin.

The use of actual photographs in Figure 1 is good.  It is hoped that the journal will publish them large enough to be viewed as intended.

The abstract should define Ir-Imes.

Author Response

Dear Editor of Molecules,

I am pleased to submit the revised version of our work entitled “A controlled System for Parahydrogen Hyperpolarization experiments”.

Manuscript ID: molecules-3949858

First, we would like to express our sincere gratitude to you and the esteemed reviewers for your efforts and valuable comments. We truly appreciate the insightful feedback provided, which has greatly helped in enhancing the quality of our work. We hope that our revisions and responses meet your expectations and address all concerns satisfactorily.

Some of the figures include material adapted from our recent publication: Mamone, F. Floreani, A. M. Faramawy, C. Graiff, L. Franco, M. Ruzzi, C. Tubaro and G. Stevanato, Phys. Chem. Chem. Phys., 2025, Advance Article, DOI: 10.1039/D5CP01773D.

We have been granted permission for reuse and have included in the caption of the relevant Figures the line: Adapted from ref. [60] with permission from the Royal Society of Chemistry.

Below our answers

Reviewer 3:

[Comment]:  The paper describes a home-made device for creating parahydrogen and illustrates it use with example spectra.

The paper is written for researchers familiar with using parahydrogen, so some steps are not described in a way that a beginner could actually perform an experiment – especially the details of handling the NMR tube at pressure and transferring the sample to the spectrometer. 

[Response]: We thank the reviewer and take this opportunity to further clarify this important point. In the Material and Methods section we have added the following:

In SABRE-SHEATH experiments, the sample is manually transferred from the mu-metal shield to the benchtop NMR in approximately 3 seconds. The NMR tube and the gas distribution system function as a single integrated unit. The tube can be lifted simply by raising the connected 4 mm o.d. polyurethane or nylon lines, which securely hold it in place through the push-in fitting connections. We verified that the PTFE seal between the NMR tube and the Y-push-in fitting (see Fig. 1a) can withstand pressures up to 8 bar, although all our experiments were conducted at 6 bar or below. Over the course of more than 200 experiments, we observed only one instance of the NMR tube detaching from the Y-fitting at 6 bar pressure. After the controlled bubbling step, the pressure can be released through the exhaust line, and the tube was manually lifted and placed inside the benchtop NMR for signal detection. In the high-field experiments, the NMR tube—connected to the parahydrogen distribution system—is positioned at high magnetic field and remains there throughout the experiment. The length of the gas lines connecting the tube to the distribution system can be adjusted as needed (approximately 4 meters in our setup), depending on the spectrometer used, to ensure that the distribution system remains outside the 5 Gauss safety line.

[Comment]:  For the intended audience this is probably OK, but for the non-expert reading this journal to try to understand whether this system should be reproduced in their lab, a few general guidelines about the importance of magnetic field shielding, etc., would enrich the paper.

[Response]: We thank the reviewer for this clearly very useful comment. In section 2.3 we have added the following lines (from 261 to 269 in the current version):

To perform SABRE-SHEATH hyperpolarization experiments on heteronuclei (e.g., ¹³C, ¹⁵N, etc.), the magnetic field during bubbling must be accurately adjusted to sub-microtesla levels, well below the Earth’s magnetic field. In practice, this can be achieved by shielding the bubbling region from external magnetic fields using a metal enclosure, inside which a solenoid coil (Fig. 3a) is installed and driven with a DC current to generate the desired longitudinal magnetic field. For completeness, we note that for 1H SABRE-SHEATH experiments the field requirements (usually around 6 mT) are less stringent and might not require a metal shield. In the proposed set up, concentric with the μ-metal shield are the solenoid coil, which generates the desired magnetic field, and the Wilmad dewar.

[Comment]:  Since Bruker NMR spectrometers are very common, it is not only acceptable but useful to provide detailed information about interfacing with the triggers from and providing code for Topspin.

[Response]: Indeed that was our goal: to ease the transition to high field controlled-parahydrogen experiments for Bruker’s users.

[Comment]: The use of actual photographs in Figure 1 is good.  It is hoped that the journal will publish them large enough to be viewed as intended.

[Comment]: The abstract should define Ir-Imes.

[Response]: We thank the reviewer very much for the appreciation of our work. We have introduced the scientific nomenclature Chloro(1,5-cyclooctadiene)[1,3-bis(2,4,6-trimethylphenyl)imidazole-2-ylidene]iridium(I) for Ir-IMes

Round 2

Reviewer 1 Report

Comments and Suggestions for Authors

The authors satisfactorily addressed many of my previous comments, although I believe that there is still a big room for improvement.

However, comment #5 about IDs (inside diameters) and ODs (outside diameters) of NMR and PTFE tubes remained unsatisfactorily addressed. The authors added Section 4.3, but I could not find information about IDs of either standard Wilmad NMR or PTFE tubes. Appendix 1 does not contain information about it as well. There are also no manufacturer catalogue numbers for PTFE and Wilmad NMR tubes, which could be used to trace the IDs.

Could the authors add information about IDs and/or provide manufacturer numbers?

At the same time, I do not think I need to read this manuscript again.

Author Response

[REVIEWER] The authors satisfactorily addressed many of my previous comments, although I believe that there is still a big room for improvement.

However, comment #5 about IDs (inside diameters) and ODs (outside diameters) of NMR and PTFE tubes remained unsatisfactorily addressed. The authors added Section 4.3, but I could not find information about IDs of either standard Wilmad NMR or PTFE tubes. Appendix 1 does not contain information about it as well. There are also no manufacturer catalogue numbers for PTFE and Wilmad NMR tubes, which could be used to trace the IDs.

Could the authors add information about IDs and/or provide manufacturer numbers?

[RESPONSE]: 

We thank the reviewer for the positive evaluation of our work. We have now added, in Appendix 1, the catalogue code of the Wilmad economy NMR tubes used, and in Section 4.3 we have also included the link to the specific PTFE tubing employed in the setup. In addition, we have provided the Zenodo DOI of our data repository for full traceability.

We appreciate the reviewer’s thorough assessment and believe that the revised manuscript now fully addresses all comments.

At the same time, I do not think I need to read this manuscript again.